# Position: Towards Bidirectional Human-AI Alignment

**Hua Shen**[1A] * **Tiffany Knearem**[2B] **Reshmi Ghosh**[3B]
**Kenan Alkiek**[4C] **Kundan Krishna**[5C] **Yachuan Liu**[4C] **Ziqiao Ma**[4C] **Savvas Petridis**[9C]
**Yi-Hao Peng**[7C] **Li Qiwei**[4C] **Sushrita Rakshit**[4C] **Chenglei Si**[8C] **Yutong Xie**[4C]
**Jeffrey P. Bigham**[7D] **Frank Bentley**[6D] **Joyce Chai**[4D] **Zachary Lipton**[7D]
**Qiaozhu Mei**[4D] **Rada Mihalcea**[4D] **Michael Terry**[9D] **Diyi Yang**[8D]
**Meredith Ringel Morris**[9E] **Paul Resnick**[4E] **David Jurgens**[4E]
[1] NYU Shanghai, New York University, [2] MBZUAI, [3] Microsoft, [4] University of Michigan,
[5] Apple, [6] Google, [7] Carnegie Mellon University, [8] Stanford University, [9] Google DeepMind
[A] Project Lead, [B] Team Leads, [C] Team Members (Equal Contributions),
[D] Advisors (Equal Contributions), [E] Lead Advisors (Equal Contributions),
huashen@nyu.edu

## Abstract

Recent advances in general-purpose AI underscore the urgent need to align AI systems with human goals and values. Yet, the lack of a clear, shared understanding of what constitutes "alignment" limits meaningful progress and cross-disciplinary collaboration. In this position paper, we argue that the research community should explicitly define and critically reflect on "alignment" to account for the bidirectional and dynamic relationship between humans and AI. Through a systematic review of over 400 papers spanning HCI, NLP, ML, and more, we examine how alignment is currently defined and operationalized. Building on this analysis, we introduce the Bidirectional Human-AI Alignment framework, which not only incorporates traditional efforts to align AI with human values but also introduces the critical, underexplored dimension of aligning humans with AI – supporting cognitive, behavioral, and societal adaptation to rapidly advancing AI technologies. Our findings reveal significant gaps in current literature, especially in long-term interaction design, human value modeling, and mutual understanding. We conclude with three central challenges and actionable recommendations to guide future research toward more nuanced, reciprocal, and human-AI alignment approaches.

## 1 Introduction

Artificial Intelligence (AI), particularly generative AI, has demonstrated remarkable capabilities in reasoning, language understanding, problem solving, and more [1]. However, its increasing integration into society raises significant risks, such as amplifying biases in hiring [2] or perpetuating stereotypes in text-to-image models [3]. These concerns highlight the urgent need to align these systems with values, ethical principles, and the goals of individuals and society at large. This need, commonly referred to as "AI alignment," [4, 5] is crucial for ensuring that AI systems function in a manner that is not only effective but also consistent with human values, minimizing harm and maximizing societal benefits. Yet, key challenges remain:

**Challenge 1: Specification Gaming.** AI designers often define objectives or feedback to align systems with human goals, but these rarely capture all intended values [6]. This leads to reliance on

---

*We denote all authors' roles, affiliations, and contributions in Appendix 13.1. This work was supported in part by the National Science Foundation under Grant No. IIS-2143529 and No. IIS-1949634. Corresponding author: Hua Shen, Assistant Professor of NYU Shanghai, New York University (huashen@nyu.edu).

39th Conference on Neural Information Processing Systems (NeurIPS 2025) Position Paper Track.

proxies like human approval [4], enabling specification gaming [7, 8], where AI makes seemingly "correct" decisions for the wrong, opaque reasons [9, 10, 11].

**Challenge 2: Scalable Oversight.** As AI systems become more complex—potentially reaching AGI [12]—aligning them through human feedback grows harder. Evaluating their behavior is often slow or infeasible [5], prompting research into reducing supervision burdens and enhancing human oversight, a challenge known as Scalable Oversight [13].

**Challenge 3: Dynamic Nature.** As AI advances, alignment must adapt to evolving human values. Without considering long-term cognitive and social impacts, AI may become neither humane nor desirable [14]. This needs a dynamic, ongoing alignment process with cross-disciplinary collaboration.

Traditionally, AI alignment has been approached as a static, one-way process focused on shaping AI to achieve desired outcomes and avoid harm [15, 1, 16]. However, this unidirectional view is increasingly insufficient as AI systems become more integrated into daily life and assume complex decision-making roles [17]. Their interactions with humans create evolving feedback loops that influence both AI behavior and human responses [18, 19, 20], highlighting the need for a more dynamic and reciprocal understanding of alignment [17].

**In this position paper, we argue that it is critical for the research community to explicitly reflect on what we mean by "alignment" and to take into account the bidirectional, dynamic interactions between humans and AI to achieve responsible and safe AI systems.**

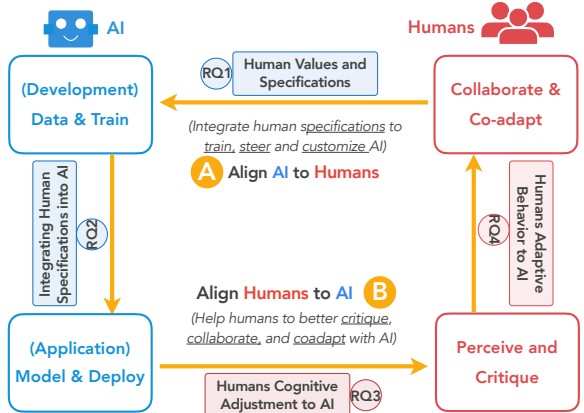

Figure 1: The overview of the Bidirectional Human-AI Alignment framework. Our framework encompasses both (A) conventional studies which focus on "Align AI with Humans" and (B) "Align Humans with AI". We further identify four key Research Questions to facilitate this holistic loop of "bidirectional human-AI alignment", and provide answers that can potentially address `RQ1-RQ4` in Section 3.

Through a systematic review of over 400 papers across HCI, NLP, ML, and related fields, we examine how alignment is currently defined and implemented. Based on this analysis, we propose the Bidirectional Human-AI Alignment (in Figure 1) framework. It extends the traditional focus on "Aligning AI to Humans"—integrating human input to train, steer, and customize AI—by introducing the equally vital yet underexplored direction of "Aligning Humans to AI", which emphasizes cognitive, behavioral, and societal adaptation to rapid AI advancement.

Our findings reveal key gaps in existing research, particularly in human value modeling, oversight of model inference, critical evaluation of AI's embedded values, and its broader societal impact. We conclude by outlining near- and long-term risks and opportunities, offering actionable recommendations to advance more reciprocal, adaptive, and nuanced approaches to human-AI alignment.

## 2 Defining Alignment: Fundamentals

Building on our analysis of systematic review (see details in Appendix 8), we explicitly identify the key definitions in alignment research and formally propose "Bidirectional Human-AI Alignment".

**Goals.** AI alignment research proposes multiple alignment goals [21, 22], such as *intentions* [1, 23], *preferences* [24, 25], *instructions* [26, 27], and *values* [28, 29]. But these terms are often used interchangeably without clear distinctions. Philosophical analysis suggests human values (moral beliefs/principles) are the most suitable alignment goal, as they ensure AI acts ethically while minimizing risks [29, 30]. Though trade-offs exist, this work adopts "human values" as the alignment objective, meaning AI should behave as individuals or society morally expect (See details in Table 2).

**Align with Whom.** AI alignment involves multiple stakeholders, such as end users [31], AI practitioners [32, 23, 33], and organizations [34]. Many studies reference "general humans" without accounting for group differences, despite the fact that stakeholders often hold conflicting values [29].

Table 1: The fine-grained typology of two directions in the Bidirectional Human-AI Alignment.

| | Research Question | Sub-Research Question | Dimensions | References |
|---|---|---|---|---|
| **Align AI with Humans** | **RQ1: Human Values and Specifications** | **Categorizing Aligned Human Values** *what values have been aligned with AI?* | Source of Values | [41, 42, 43] |
| | | | Value Types | [16, 44, 45] |
| | | **Interaction Techniques to Specify AI Values** *How humans could interactively specify values in AI development?* | Explicit Human Feedback | [46, 47, 26] |
| | | | Implicit Human Feedback | [48, 49, 50] |
| | | | Simulated Human Value Feedback | [51, 52, 53] |
| | **RQ2: Integrating Human Specification into AI** | **Develop AI with General Values** *how to incorporate general human values into AI development?* | Instruction Data | [54, 55, 56] |
| | | | Model Learning | [1, 57, 58] |
| | | | Inference Stage | [26, 59, 60] |
| | | **Customizing AI for Individuals or Groups** *how to customize AI to incorporate values of individuals or groups?* | Customized Data | [61, 62, 63] |
| | | | Adapt Model by Learning | [64, 65, 66] |
| | | | Interactive Alignment | [67, 68, 69] |
| | | **Evaluating AI Systems** *how to evaluate AI regarding human values?* | Human-in-the-loop Evaluation | [70, 71, 43] |
| | | | Automatic Evaluation | [72, 55, 73] |
| | | **Ecosystem** *how to build the ecosystem to facilitate human-AI alignment?* | Platforms | [56, 74, 75, 76, 77, 78] |
| **Align Humans with AI** | **RQ3: Human Cognitive Adjustment to AI** | **Perceiving and Understanding of AI** *how do humans learn to perceive and explain AI systems?* | Education and Training Human | [79, 80, 81] |
| | | | AI Sensemaking and Explanations | [82, 83, 84] |
| | | **Critical Thinking about AI** *how do humans think critically about AI systems?* | Trust and Reliance on AI Decisions | [85, 86, 87] |
| | | | Ethical Concerns and AI Auditing | [88, 89] |
| | | | Calibrate Cognition to Align AI | [90, 91, 92] |
| | **RQ4: Human Adaptation Behavior to AI** | **Human Collaborating with Diverse AI Roles** *how do humans collaborate with AI with differing capability levels?* | AI Assistants | [93, 94, 95] |
| | | | AI Partners | [21, 96, 97] |
| | | | AI Tutors | [98, 99, 100] |
| | | **AI Impacts on Humans and Society** *how are humans influenced by AI systems?* | Impact on Individual Behavior | [18, 101] |
| | | | Societal Concerns and AI Impacts | [102, 103] |
| | | | Reaction to AI Advancements | [104, 105] |
| | | **Evaluation in Human Studies** *how might we evaluate the impact of AI on humans and society?* | Evaluate Human-AI Collaboration | [106, 107] |
| | | | Evaluate Societal Impact | [16, 108] |

Rather than pursuing a universal moral theory, prior work advocates aligning AI with principles tailored to compatible human groups [28]. *Pluralistic value alignment*, grounded in social choice theory [35, 28], provides a framework for integrating diverse perspectives [28]. We adopt this view, recognizing that aligning with affected individuals and groups entails ongoing challenges.

**Align with What.** While prior studies have sought to align AI with human values, these values are often vaguely or inconsistently defined. To address this, we reviewed several prominent value theories, including Moral Foundations Theory [36] and Social Norms and Ethics [37], drawing from psychology and social science. We selected the Schwartz Theory of Basic Values [38, 39] for its demonstrated cross-cultural applicability, relevance across contexts (individuals, interactions, groups), and frequent adoption in NLP research [40, 41]. This framework consists of eleven types of universal values, and defines values as beliefs about desirable end states or behaviors that transcend specific situations and guide evaluation and decision-making.

*Definition*: **Bidirectional Human-AI Alignment** is a comprehensive framework that encompasses two interconnected alignment processes: 'Aligning AI with Humans' and 'Aligning Humans with AI'. The former focuses on integrating human specifications into training, steering, and customizing AI. The latter supports human agency, empowering people to think critically when using AI, collaborate effectively with it, and adapt societal approaches to maximize its benefits for humanity.

## 3    Bidirectional Human-AI Alignment Framework

This section introduces the Bidirectional Human-AI Alignment framework, which encompasses two interconnected alignment directions as a feedback loop, as shown in Figure 1. The "**Align AI to Humans**" direction refers to mechanisms to ensure that AI systems' values match those of humans'. The "**Align Humans to AI**" direction investigates human cognitive and behavioral adaptation to AI advancement. We introduce more details below.

### 3.1    Align AI to Humans

This direction delineates alignment research from **AI-centered perspective** (e.g., ML/NLP domains) and provides **AI developers and researchers** with approaches for handling two main challenges: carefully specifying the values of the system, and ensuring that the system adopts the specification robustly [4, 33]. Therefore, as shown in Table 1, we explore two core research questions in this direction as: **RQ1.** *What relevant human values are studied for AI alignment, and how do humans specify these values?* and **RQ2.** *How can human values be integrated into the AI systems?*

• **RQ1: Human Values and Specifications.**

To identify key human values and specifications for AI alignment, we begin by addressing two critical subquestions: (1) what values have AI systems been aligned with? and (2) how can humans

interactively specify values during AI development? As summarized at the top of Table 1, we present the key dimensions that emerged from our analysis of these questions, along with supporting studies.

**Categorizing Aligned Human Values.** To systematically understand human values relevant to human-AI alignment [109, 110], we draw on the adapted Schwartz Theory of Basic Values, examining values along two dimensions: Sources and Types. The **Sources** dimension categorizes values as individual (e.g., personal interests and biological needs like factuality or cognitive biases) [38, 41, 42], social (e.g., shared group norms such as fairness or morality) [38, 43, 45], and interactive (e.g., expectations in human-AI interactions like usability, autonomy, and trust) [111, 112, 113]. The **Types** dimension organizes values into four high-order categories: Self-Enhancement (e.g., achievement, power) [114, 90, 115], Self-Transcendence (e.g., benevolence, honesty, fairness) [116, 117, 118], Conservation (e.g., safety, tradition, conformity) [119, 31, 96], Openness to Change (e.g., creativity, privacy, autonomy) [120, 121, 122]. These dimensions offer a comprehensive framework for evaluating and aligning AI systems with the multifaceted nature of human values.

**Interaction Techniques to Specify AI Values.** This sub-research question explores how human values are interactively specified to ensure AI alignment, focusing on the techniques through which AI systems manifest or internalize these values. It identifies three main approaches: **explicit human feedback**, where values are directly communicated via principles, ratings, natural language interactions, or multimodal inputs like gestures and images [46, 47, 26]; **implicit human feedback**, where values are inferred from indirect cues such as discarded options, language patterns, theory of mind reasoning, and social relationships [48, 49, 50]; and **simulated human value feedback**, where AI systems approximate human responses using feedback simulators, comparisons to human data, or synthetically generated data [51, 52, 53]. Together, these approaches illuminate the mechanisms by which AI systems interpret and enact human values via direct and indirect human-AI interaction.

*Key Takeaways.* By comparing prior research with our comprehensive analysis of human values and interaction techniques, we found that existing studies are largely constrained to conventional principles and standard interaction methods, overlooking the broader spectrum of human values and the interactive approaches needed to specify them effectively in alignment.

- **RQ2: Integrating Human Specifications into AI.**

Building on the value-laden human specifications gathered through interaction, we next explore diverse methods for integrating human values into AI systems. We examine this central question across two key stages of the AI lifecycle—development and deployment—by asking: (1) how can general or customized human values be integrated throughout the AI development process? and (2) what methods and platforms are available to evaluate values during AI development?

**Integrating General Values to AI.** This sub-research question examines how broad, universally recognized human values are embedded into AI systems to ensure ethical alignment and societal acceptance. It outlines three key dimensions: **Instruction Data**, which includes human annotations, human-AI co-annotation, and simulated human data to guide value-based training [54, 55, 56]; **Model Learning**, where human values are integrated during training through either real-time online alignment or offline processes prior to deployment [1, 57, 58]; and **Inference Stage**, where AI systems are evaluated and refined using techniques such as prompting, external tool interactions, and response search to ensure their outputs align with predefined ethical criteria [26, 113, 59, 60]. Together, these processes aim to build AI systems that promote trust and responsible use.

**Customizing AI Values.** This sub-research question investigates how AI systems can be customized to reflect specific user preferences, application domains, or community values, thereby improving contextual alignment. It identifies three primary strategies: **Customized Data**, which involves curating and finetuning datasets based on socio-demographic groups, user histories, or expert selections to align models with targeted human values [61, 62, 63]; **Adapt Model by Learning**, which includes techniques such as group-based learning, active learning, adapter insertion, mixture of experts, and enhanced knowledge integration to refine model behavior [64, 65, 66]; and **Interactive Alignment**, which actively engages users through real-time feedback, steering prompts, and proactive adjustments based on user profiles to tailor AI systems to specific contexts and preferences [67, 68, 69].

**Evaluating AI Systems.** This sub-research question examines how the integration of human values into AI systems, particularly large language models (LLMs), is evaluated, highlighting both human-in-the-loop and automated methods. **Human-in-the-loop evaluation** involves human judgment, feedback, and collaboration to assess the ethical and value alignment of AI outputs, leveraging direct

human input or combined human-AI assessment processes [70, 71, 43]. **Automatic evaluation** utilizes computational techniques, including human simulators, standardized benchmarks, and distributional comparisons, to evaluate alignment without human intervention [72, 55, 73]. Together, these approaches aim to ensure that AI systems reflect human ethical standards and value frameworks effectively and reliably.

**Ecosystem and Platforms.** The ecosystem and platforms refer to the broader context in which AI systems operate and interact with other agents, platforms, or environments. This includes the infrastructure, frameworks, and technologies that support the development, deployment, and utilization of AI systems. **LLM-based Agents** are based on large language models (LLMs) such as GPT (Generative Pre-trained Transformer) models, which have been pre-trained on vast amounts of text data [56, 74, 123, 75]. **RL-based Agents** are based on reinforcement learning (RL) algorithms to learn and adapt their behavior based on feedback from the environment or human users [76]. **Annotation Platforms** refers to the ecosystems that are designed to crowdsource human demonstrations as collected data for reinforcement learning [77] and supervised finetuning learning for alignment [78].

*Key Takeaways.* Our systematic analysis of value integration and evaluation in AI reveals a strong focus on explicit value annotations, while implicit value expressions and behaviors are often neglected. Despite the power of general-purpose AI, methods for customizing systems to reflect individual or group values remain underexplored. Additionally, there is a lack of standardized criteria for evaluating human-in-the-loop methods and supporting platforms, underscoring the need for more robust and context-sensitive evaluation frameworks in value-aligned AI development.

## 3.2 Align Humans to AI

From a *long-term* perspective, it is crucial to consider the dynamic and evolving nature of human-AI alignment. This direction emphasizes a **human-centered perspective**—drawing from fields such as HCI and the social sciences—and offers guidance for **researchers and user experience designers** in addressing two core research questions: **RQ3.** *How can humans learn to perceive, explain, and critique AI?* and **RQ4.** *How do individuals and society adapt their behaviors in response to AI advancements?*

- **RQ3: Human Cognitive Adjustment to AI.**

For effective collaboration and value specification, humans need to develop a clear understanding of how AI systems function. As AI introduces various risks, fostering critical thinking is essential to prevent blind reliance. To address this, we systematically investigate, as summarized in Table 1, (1) how humans learn to perceive and understand AI, and (2) how they can engage in critical reflection on AI behavior and outputs.

**Perceiving and Understanding AI.** This sub-research question explores how to enhance human understanding and perception of AI systems, particularly among non-technical users, through education, training, and human-centered explanation techniques. It emphasizes the importance of **AI literacy and awareness** as foundational competencies for effective human-AI collaboration [79], supported by **explicit training courses** designed to improve users' ability to engage with AI [80, 81]. Additionally, it highlights efforts in **AI sensemaking and human-centered explanations**, including visualizations and interactive techniques, to help people interpret AI mechanisms and outputs, thereby fostering more informed and meaningful human-AI interactions [82, 83, 84].

**Critical Thinking around AI.** This sub-research question examines how individuals critically reflect on and evaluate AI systems by comparing their own mental models with those of the AI, focusing on the rationality, reliability, and ethical behavior of these technologies. It emphasizes the need for humans to develop critical thinking skills to identify biases, errors, and ethical concerns in AI outputs, and to audit AI systems for compliance with moral and societal standards. Key areas include building appropriate levels of **Trust and Reliance on AI** based on its competence and reliability [85, 86, 87, 124], engaging in **selective AI adoption** aligned with user needs and values [125, 126], and addressing **Ethical Considerations** through mitigation strategies and auditing [88, 89]. Additionally, it underscores the importance of **Recalibrating Cognition**, where users adjust their understanding and expectations of AI performance and reliability to foster a balanced and informed relationship with AI systems [90, 91, 92].

*Key Takeaways.* Our systematic review reveals a significant gap in research on training and educating humans to develop appropriate knowledge, trust, and reliance when collaborating with AI systems.

Additionally, there is limited exploration of how to critically evaluate AI across a broad spectrum of human values.

- **RQ4: Human Adaptation Behavior to AI.**

Building on our investigation of human cognitive adjustments to AI, we further examine how individuals and society can respond effectively to AI's expanding influence. To guide this inquiry, we address three key questions: (1) How do humans learn to collaborate with AI across its diverse roles? (2) In what ways are individuals and society affected by AI? and (3) How can these impacts be comprehensively assessed? The key dimensions emerging from this analysis are summarized at the bottom of Table 1.

**Human-AI Collaboration Mechanisms.** This category explores the diverse ways humans and AI collaborate through partnerships, co-creation, and mutual learning. **AI Assistants**, particularly those powered by LLMs, support human tasks by interpreting user demands, enhancing prompt formulation, generating creative prototypes, and aiding decision-making [93, 94, 95]. In collaborative frameworks, humans and AI function as **AI Partners**, where simulated agency and reciprocal learning enable joint decision-making and knowledge sharing [21, 96, 97]. AI delegation and mediation transform traditional human tasks, while co-design with AI treats the system as both a collaborator and a design material. Additionally, **AI Tutoring** systems enhance human learning in both technical and social domains by offering tailored feedback, adaptive instruction, and immersive practice environments, ultimately improving skill acquisition and performance [98, 99, 100].

**AI Impact on Humans and Society.** This category investigates the multifaceted effects of AI advancement on human behavior, attitudes, and societal dynamics, aiming to inform policy, education, and interventions. The **Impacts on Participatory Individuals and Groups** dimension focuses on how AI influences human decision-making, creativity, privacy, and authorship—shaping behaviors and raising concerns about data rights and intellectual ownership [18, 101, 127, 128]. The **Societal Concerns and AI Impacts** dimension expands this lens to the societal level, examining AI's effects on misinformation, education, social norms, and the workplace, including issues like disinformation, shifts in learning practices, changes in interpersonal relationships, and job displacement [102, 103, 129]. The **Reaction to AI Advancement** dimension explores regulatory, cultural, and institutional responses to AI, addressing how societies perceive, govern, and adapt to AI technologies [104, 105, 130]. This includes efforts to regulate bias and discrimination, develop policy frameworks, track evolving AI acceptance, ensure transparency and oversight, and establish responsible AI checklists to guide ethical and safe deployment.

**Evaluation in Human Studies.** This summary covers common empirical methods used to rigorously evaluate AI's impact on humans at both micro and macro levels. At the micro-level, **Human-AI Collaboration Evaluation** assesses not only task success and efficiency but also user experience, including cognitive workload, user satisfaction, control, and trust, especially in critical settings to prevent failures [106, 131, 107, 106]. Methods include quantitative interaction analytics, qualitative surveys and interviews, and statistical analyses to understand and verify user behaviors and perceptions. At the macro-level, **Societal Impact Evaluation** focuses on understanding long-term behavioral changes within large populations as AI use becomes widespread, employing large-scale public opinion surveys and behavioral data analytics over time to capture evolving patterns and societal shifts related to AI interaction [16, 108, 132].

*Key Takeaways.* Our review reveals that while prior studies have extensively examined how AI can assist humans across various tasks, there is limited exploration of the challenges humans face when collaborating with AI that surpass human capabilities in certain domains. Additionally, more research is needed to understand the evolving impact of AI on individuals and society at large over time.

## 4 Underexplored Research Gaps and Challenges

In this section, we consolidate key findings from our systematic analysis of the framework and the reviewed literature (in Figure 2. To accurately capture the distribution of existing research and identify current gaps and challenges, we quantified the number of relevant studies corresponding to each dimension in the Bidirectional Human-AI Alignment. We elaborate on key findings below.

**Underexplored Dimensions in Aligning AI with Humans.** Current AI alignment research has primarily focused on incorporating explicitly stated human values, often gathered through direct feedback mechanisms such as ratings, rankings, or instructions. However, several critical dimensions remain underexplored. First, the **use of implicit human feedback** — such as behavioral cues,

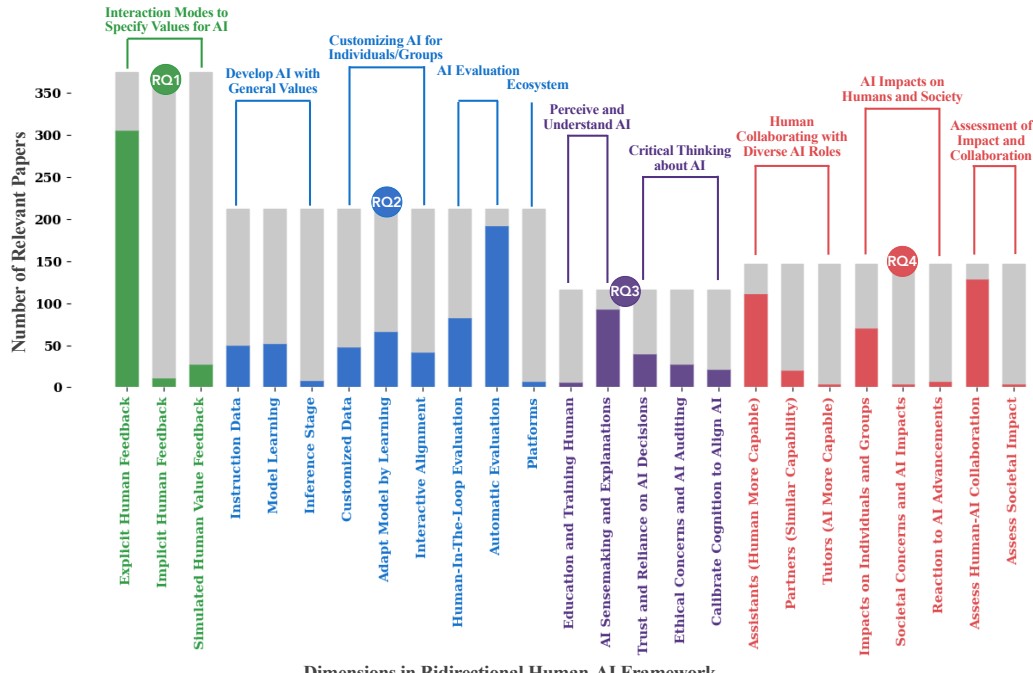

Figure 2: The number of papers for each dimension in the *bidirectional human-AI alignment* framework. Out of papers that are relevant to each research question (*i.e.,* gray bars), we show the number of papers that are relevant to each dimension (*i.e.,* color bars).

physiological signals, or interaction patterns—and simulated human value feedback has received limited attention, despite their potential to provide rich, context-sensitive information about human values. *Second*, most alignment efforts concentrate on model training stages, whereas **developing and customizing AI models** during inference or through interactive processes for embedding and evaluation values remains significantly underexplored. Enabling real-time adaptation of AI behavior to human input could enhance alignment in dynamic or personalized contexts. *Third*, **human-in-the-loop evaluation**, which involves assessing AI systems through active human participation and feedback, is rarely used in comparison to fully automated evaluation metrics. Expanding research into these areas is essential for advancing more robust, responsive, and context-aware alignment.

**Underexplored Dimensions in Aligning Humans with AI.** Human-centered alignment research has predominantly emphasized designing AI systems that facilitate human understanding through sensemaking and explanation—primarily by clarifying the justifications behind AI decisions to foster user trust and reliance. However, this focus often **overlooks the broader goal of fostering AI literacy**—the essential skills and competencies individuals need to understand, critique, use, and interact effectively with AI systems. Despite its foundational role in responsible AI engagement, AI literacy remains an underexplored area. Furthermore, while numerous studies have proposed interactive mechanisms and prototypes to support human-AI collaboration, they commonly assume that AI operates in a subordinate or assistive role. As AI systems grow increasingly capable, research must also **consider collaborative dynamics between humans and AI** with equal or superior capabilities. Additionally, the **ethical auditing** of AI from a human-centered perspective and the **societal-level impacts of AI** — such as changes in human behavior, social relationships, and public responses—have not been sufficiently examined. These dimensions are critical for ensuring meaningful and equitable alignment between humans and evolving AI technologies.

## 5  Near to Long-term Risks and Opportunities

Drawing upon insights gained from the development of our framework and the associated systematic review analysis, we propose future research aiming to achieve the long-term alignment goal by identifying three important challenges from near-term to long-term objectives, including the Specification Game, Dynamic Co-evolution of Alignment, and Safeguarding Coadaptation.

## 5.1 Specification Game

An important near-term challenge is resolving the "Specification Game", which involves precisely defining and implementing AI goals and behaviors to align with human intentions and values. Next, we will introduce how synergistic efforts from two directions can potentially address this challenge.

**Integrate fully specified human values into aligning AI.** Individuals often possess value systems that encompass multiple values with varying priorities, rather than a single value, to guide their behaviors [38, 39]. Also, these priorities can change dynamically throughout an individual's life stages. As such, It is more realistic to select values compatible with specific societies or situations, given the fact that we live in a diverse world [29]. Future research, inspired by Social Choice Theory [35], could focus on using democratic processes to aggregate individual values into collective agreements. Building on the summaries in Sections 3.1, researchers can employ democratic methods to identify diverse subsets of human values for AI alignment. Additionally, creating datasets that represent these values is crucial. Besides, it is crucial yet challenging *for AI designers to investigate how to fully specify the appropriate values and to further integrate these values into AI alignment.* Future important area involves developing algorithms, such as the Bradley-Terry Model [58] or Elo Rating System [26], to convert heterogeneous human values into AI-compatible formats for training reward models and guiding reinforcement learning. Researchers should also explore AI models capable of aligning with unstructured human data, including free-form descriptions of values, multimedia, or sensor recordings depicting human behavior.

**Elicit nuanced and contextual human values during diverse interactions.** Current alignment methods use instructions, ratings, and rankings to infer human values, which can not fully capture all relevant human values and constraints. Future research should focus on optimizing interactive interfaces to efficiently elicit human values. These interfaces can leverage diverse interaction modes to capture comprehensive human value information. Additionally, people often struggle to formulate optimal prompts for AI, accurately specify their requirements, and articulate their desired values, which can change based on context and time. Developing proactive interfaces that use conversational techniques to elicit nuanced and evolving values is also crucial. Implicit human signals that indicate values are also frequently overlooked. Additionally, systems that track interactions to hypothesize and validate implicit human values in real-time should be designed.

## 5.2 Dynamic Co-evolution of Alignment

The challenge ahead lies in comprehending and effectively navigating the dynamic interplay among human values, societal evolution, and the progression of AI technologies. Future studies in these directions aim to bolster a synergistic co-evolution between AI and human societies, adapting both to each other's changes and advancements.

**Co-evolve AI with changes in humans and society.** Existing literature often treats AI alignment as static, ignoring its dynamic nature. A long-term perspective must consider the co-evolution of AI, humans, and society. As AI systems evolve and scale up, they gain new capabilities, making it essential to ensure their goals remain aligned with human values. Thus, alignment solutions require continuous oversight and updates. Future research should develop methods for continuously updating AI with limited data without compromising alignment values and performance. This could involve forecasting human value evolution and preparing AI with flexible strategies like prompting or interventions. *(ii)* Additionally, AI advancements also influence human actions and values, necessitating adaptive alignment solutions. Ensuring AI co-evolves with human and societal changes is crucial for robust alignment. This challenge could potentially be addressed by forecasting the potential evolution trajectories of human values or behavioral patterns, and preparing AI with the flexibility to adapt in advance, for example, through prompting or intervention strategies.

**Adapt humans and society to the latest AI advancements.** While current AI systems lag behind humans in many tasks, identifying and handling AI mistakes, including knowing when to seek human intervention, remains essential. Future research should focus on developing validation mechanisms that enable humans to interpret and verify AI outputs. This could involve designing interfaces that allow humans to request step-by-step justifications from AI or integrating tools to verify the truthfulness of AI referring to Section 3.2. Additionally, developing interfaces that enable groups of humans to collaboratively validate AI outputs and creating scalable validation tools for large-scale applications are important directions. *(ii)* As AI advances, it becomes essential to develop systems that enable humans to utilize AI with capabilities surpassing their own. Research is needed to understand how individuals can interpret and validate AI outputs for tasks beyond their abilities and

leverage advanced AI sustainably, avoiding issues like job displacement or loss of purpose. Another research direction is designing strategies to enhance human capabilities by learning from advanced AI, including gaining knowledge and building skills. *(iii)* As AI integrates more into daily tasks, its impact on human values, behaviors, capabilities, and society remains uncertain. Continuous examination of AI's influence on individuals, social relationships, and broader societal changes is vital. Research should assess how humans and society adapt to AI advancements, guiding AI's future evolution. Potential areas include evaluating changes in individual behavior, social relationships, and societal governance as AI replaces traditional human skills. Understanding these dynamic changes is essential for grasping the broader impact of AI on humanity and society.

### 5.3 Safeguarding Co-adaptation

As AI gains autonomy and capability, the risks associated with its instrumental actions, as a means toward accomplishing its final goals, increase. These actions can be undesirable for humans. Therefore, safeguarding the co-adaptation between humans and AI is crucial. We next explore future research to address this challenge from both directions.

**Specify the goals of an AI system into interpretable and controllable instrumental actions for humans.** As advanced AI systems become more complex, they present greater challenges for human interpretation and control. It is crucial to empower humans to detect and interpret AI misconduct and enable human intervention to prevent power-seeking AI behavior. Research should focus on designing corrigible mechanisms for easy intervention and correction, including modular AI architectures and robust override protocols that allow human operators to halt or redirect AI activities. These components should be human-interpretable, enabling scenario testing. *(ii)* Furthermore, advanced AI systems may intentionally mislead or disobey humans, generating plausible fabrications [133]. Developing reliable interpretability mechanisms to validate the faithfulness and honesty of AI behaviors is essential. This includes correlating AI behaviors with internal neuron activity signals, akin to physiological indicators in human polygraph tests [134]. Inspecting these indicators can help humans assess the truthfulness of AI interpretations and prevent risky actions.

**Empower humans to identify and intervene in AI instrumental and final strategies in collaboration.** Preventing advanced AI from engaging in risky actions requires robust human supervision. Essential steps include developing training and simulation environments with scenario-based exercises and timely feedback, and creating interactive dashboards for real-time monitoring. These dashboards should feature effective data visualization, anomaly detection, and prompt alert systems for immediate intervention. *(ii)* Scalable solutions are needed for supervising AI across various applications. Real-time oversight becomes more challenging with widespread AI deployment, necessitating advanced autonomous monitoring tools. These tools should learn normal AI behavior and flag deviations immediately. Integrating training environments, interactive dashboards, and scalable diagnostic tools will enhance human ability to ensure better alignment with human values.

## 6 Limitations

One limitation of this work is the scope of the sampled and filtered papers. The rapidly growing literature on human-AI alignment spans diverse venues across many domains. Instead of an exhaustive collection, we focused on developing a holistic bidirectional human-AI alignment framework using essential research questions, dimensions, and codes. Our surveyed papers and team members primarily focus on computing-related fields like ML, NLP, and HCI, though alignment research also involves disciplines like cognitive science, psychology, and STS (Science, Technology, and Society). Our framework can naturally extend to these areas. Despite these limitations, we believe our bidirectional human-AI alignment framework serves as a foundational reference for future researchers.

## 7 Conclusion

This study clarifies the conceptual foundations of human-AI alignment by analyzing how key terms are defined and operationalized across over 400 papers from NLP, HCI, ML, and related domains. We introduce the Bidirectional Human-AI Alignment framework, which organizes alignment efforts into two interdependent directions: aligning AI with humans values, and aligning humans with AI – enabling humans to effectively understand, evaluate, and adapt to AI systems. Our analysis identifies critical gaps in current literature, including limited support for long-term interaction, underdeveloped models of human values, and challenges in mutual intelligibility. We conclude with three central challenges—specification gaming, scalable oversight, and dynamic alignment—and offer actionable recommendations to support future research aimed at fostering reciprocal, robust, and context-aware approaches to human-AI alignment.

# APPENDIX

## 8 Systematic Literature Review

### 8.1 Systematic Literature Review Process

To understand the research literature relevant to the ongoing, mutual process of human-AI alignment, we performed a systematic literature review based on the PRISMA guideline [135, 136]. Figure 3 shows the workflow of our process for paper coding and developing the *bidirectional human-AI alignment* framework. We introduce the step details below.

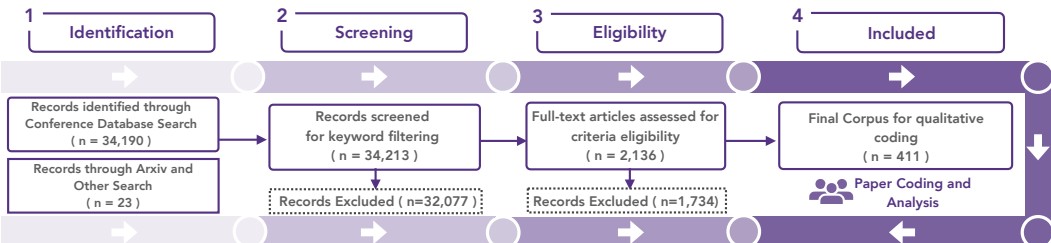

Figure 3: The selection and refinement process of our systematic literature review. We referred to the PRISMA guideline [135, 136] to report the workflow. From the identification of 34,213 records by keyword search, to screen eligible papers against our criteria and arriveg at our final corpus of 411 papers. For each of the stages where literature reviews were excluded (identification, screening, and eligibility) we further present the total of excluded records.

#### 8.1.1 Identification and Screening with Keywords.

We started with papers published in the AI-related domain venues (including NLP, HCI, and ML fields) beginning from the advent of general-purpose generative AI to present, *i.e.,* primarily between January, 2019 and January, 2024 (see details in Appendix 8.2). We retrieved 34,213 papers in the initial *Identification* stage. Further, we collectively defined a list of keywords (see details in Appendix 8.3) and screened for papers that included at least one of these keywords (*e.g.,* human, alignment) or their variations in the title or abstract. We included 2,136 papers in *Screening* stage.

#### 8.1.2 Assessing Eligibility with Criteria

We further filtered the 2,136 papers based on explicit inclusion and exclusion criteria, i.e., the *Eligibility* stage. Our criteria revolved around six research questions that we collectively identified to be most pertinent to the topic, including *1) what essential human values have been aligned by some AI models? 2) how did we effectively quantify or model human values to guide AI development? 3) what strategies have been employed to integrate human values into the AI development process? 4) how did existing studies improve human understanding and evaluation of AI alignment? 5) what are the practices for designing interfaces and interactions that facilitate human-AI collaboration? 6) How have AI been adapted to meet the needs of various human value groups?* We included papers that could potentially answer any of these questions. Further, based on the scope in Section **??**, we excluded papers that did not meet our inclusion criteria. This resulted in a final corpus of 411 papers, which were analyzed in detail using qualitative coding (see Appendix 8.4 for more details).

#### 8.1.3 Qualitative Code Development.

Referring to the code development process in [20], we first conducted qualitative coding for each paper by identifying relevant sentences that could answer the above research questions, and entering short codes to describe them into a codebook. We iteratively coded relevant sentences from each paper through a mix of inductive and deductive approaches, which allowed flexibility to expand, modify or change the driving research questions based on our learnings as we went through the process. To ensure rigor in our coding process, two authors coded each paper. The first author independently annotated all papers after reviewing the paper abstracts and introductions. Twelve team members

each annotated a subset of the paper corpus. Our corpus includes papers from different domains (*e.g.,* HCI, NLP and ML). Therefore, we divided the authors into HCI and NLP/ML[2] teams and assigned the papers accordingly based on expertise. All team members coded each of their assigned papers to answer all six questions (if applicable) introduced above.

### 8.1.4 Framework Development and Rigorous Coding.

After developing annotations, all authors collaborated to create the bidirectional human-AI alignment framework by integrating the annotations within each of the codes. The initial version of the framework was proposed by the author who reviewed all papers. This framework furthermore underwent iterative improvement through: *1)* discussions with all team members involved in paper coding, and *2)* revisions based on feedback from the project advisors. Additionally, we strengthened the framework by reviewing papers from the AI Ethics conferences (including FAccT and AIES), and related work of the collected papers that covered other domains such as psychology and social science. We further added missing codes and papers to ensure comprehensive coverage (see Appendix 8.2 for details). The final bidirectional human-AI alignment framework, with detailed topologies, is presented in Section 3. Following the framework's finalization, we conducted another separate coding process to annotate *whether each paper investigated dimensions within our framework*. Two authors independently coded each paper.[3] These codes were then used to perform quantitative and qualitative analyses, as presented in Section **??**.

## 8.2 Venues

We primarily focused on papers from the fields of HCI, NLP, and ML ranging from year 2019 to 2024 January. We included all their papers tracks (*e.g.,* CSCW Companion and Findings) without including workshops of conferences. From the ACL Anthology, OpenReview and ACM Digital Library, we retrieved 34,190 papers into a Reference Manager Tool (*i.e.,* Paperpile). Particularly, the venues we surveyed are listed below.

- **HCI**: CHI, CSCW, UIST, IUI;
- **NLP**: ACL, EMNLP, NAACL, Findings
- **ML**: ICLR, NeurIPS
- **Others**: ArXiv, FAccT, AIES, and other related work

Additionally, we also consolidate the framework by reviewing the papers published in FAccT and AIES (*i.e.,* important venues for AI Ethics research) between 2019 and 2024 and supplemented the codes, including the AI Regulatory and Policy code in Section **??** and the exemplary paper of Regulating ChatGPT [105]), which were not covered by the original collections. Also, we include a number of papers in the "Other" class are found by related work that are highly relevant to this topic.

## 8.3 Keywords

We decided on a list of keywords relevant to bidirectional human-AI alignment. The detailed keywords include:

- **Human**: Human, User, Agent, Cognition, Crowd
- **AI**: AI, Agent, Machine Learning, Neural Network, Algorithm, Model, Deep Learning, NLP
- **LLM**: Large Language Model, LLM, GPT, Generative, In-context Learning
- **Alignment**: Align, Alignment
- **Value**: Value, Principle
- **Trust**: Trust, Trustworthy
- **Interact**: Interact, Interaction, Interactive, Collaboration, Conversational

---

[2]Note that NLP and ML are two different domains, we combine them together for the purposes of literature review analysis since they both work on developing and evaluating AI technologies.

[3]The joint probability of agreement for the paper annotations was 0.78.

- **Visualize**: Visualization, Visualize
- **Explain**: Interpretability, Explain, Understand, Transparent
- **Evaluation**: Evaluate, Evaluation, Audit
- **Feedback**: Feedback
- **Ethics**: Bias, Fairness

## 8.4 Inclusion and Exclusion Criteria

To further filter the most relevant papers among the keyword-filtered 2136 papers, we identified the six most important research questions we are interested in. We primarily selected the potential papers that can potentially address these six questions after reviewing their title and abstracts. The six topics of research questions in our filtering include:

RQ.1 **[human value category]** What essential human values have been aligned by some AI models?

RQ.2 **[quantify human value]** How did we effectively quantify or model human values to guide AI development?

RQ.3 **[integrate human value into AI]** What strategies have been employed to integrate human values into the AI development process?

RQ.4 **[assess / explain AI regarding human values]** How did existing studies improve human understanding and evaluation of AI alignment?

RQ.5 **[human-AI interaction techniques]** What are the practices for designing interfaces and interactions that facilitate human-AI collaboration?

RQ.6 **[adapt AI for diverse human values]** How has AI been adapted to meet the needs of various human value groups?

Particularly, we provide elaborated inclusion and exclusion criteria during our paper selection as listed below. We are aware that we have limitations during our paper filtering process.

**Inclusion Criteria:**

- [**Human values**] we include papers that study human value definition, specification and evaluation in AI systems.
- [**AI development techniques**] We include techniques of developing AI that aim to be more consistent with human values with interactions along all AI development stages (e.g., data collection, model construction, etc.)
- [**AI evaluation, explanation and utilization**] we include papers that build human-AI interactive systems or conduct human studies to better evaluate, explain, and utilize AI systems.
- [**building dataset with human interaction**] especially responsible dataset.

**Exclusion Criteria:**

- [**Alignment not between human & AI**] we do not include alignment studies that are not between human and AI, such as entity alignment, cross-lingual alignment, cross-domain alignment, multi-modal alignment, token-environment alignment, etc.
- [**AI models beyond LLMs - Modality**] we do not focus on AI models other than LLMs (e.g., 3D models, VR/AR, voice assistant, spoken assistant), our primary model modality is text. Specifically, we do not consider audio / video data; we do not consider pure computer vision modality.
- [**No human-AI interaction**] we do not consider studies that do not involve the interaction between human and AI, such as (multi-agent) reinforcement learning. Specifically, we do not consider interactions via voices/speech, Do not consider game interaction; Do not consider interaction for Accessibility; Do not consider Mobile interaction; Not consider autonomous vehicle interaction wearable devices, or Physical interaction;
- [**Tasks**] art and design, emotion.

- [**No human included**]

- [**focus on English**] primarily focus on English as the main language;

- [**Application**] not include the NLP papers tailored for a specific traditional task, such as translation, entity recognition, sentiment analysis, knowledge graph, adversarial and defense, topic modeling, detecting AI generations, distillation, low resource, physical robots, text classification, games, image-based tasks, hate speech detection, Human Trafficking, etc.

- [**Visualizing Embeddings**] Visualizing/interacting transformer embeddings?

- [**Embedding-based**] explanation, evaluation, etc.

- [**multi-agent reinforcement learning with self-play and population play**] techniques, such as self-play (SP) or population play (PP), produce agents that overfit to their training partners and do not generalize well to humans.

We acknowledge the extensive scope and rapid advancements of research in this area, and posit that our study offers insights that can be generalized to various modalities. For example, the value taxonomy and human-in-the-loop evaluation paradigm outlined in our framework can be applied to both text-based and other modality-based (*e.g.,* vision, robotics) models. It's worth noting that our literature review does not aim to exhaustively cover all papers in the field, which is impossible given the rapid advancement of human-AI alignment research. Instead, we adopt a human-centered perspective to review more than 400 key studies in this domain, focusing on delineating the framework landscape, identifying limitations, future directions, and a roadmap to pave the way for future research.

## 9 Selected Paper List

- **Human-Centered Studies**: [137, 107, 138, 92, 139, 140, 141, 142, 143, 144, 145, 96, 146, 147, 148, 149, 150, 151, 152, 86, 117, 126, 153, 154, 155, 156, 87, 157, 158, 159, 160, 161, 162, 104, 163, 164, 165, 166, 167, 168, 169, 32, 170, 171, 172, 173, 174, 175, 176, 3, 177, 178, 179, 180, 181, 89, 182, 183, 184, 185, 186, 187, 188, 189, 190, 191, 192, 193, 194, 122, 195, 196, 85, 91, 88, 197, 198, 199, 200, 201, 202, 203, 204, 205, 206, 207, 208, 209, 210, 121, 81, 211, 83, 212, 213, 214, 215, 90, 216, 112, 217, 218, 219, 220, 221, 222, 223, 95, 224, 21, 225, 226, 227, 94, 228, 229, 230, 231, 93, 232, 233, 234, 235, 236, 237, 238, 239, 240, 241, 242, 243, 244, 245, 246, 247, 248, 249, 250, 251, 252, 253, 254, 255, 256, 79, 46, 257, 125, 137]

- **AI-Centered Studies**
  [258, 118, 78, 52, 24, 25, 259, 75, 48, 260, 261, 262, 263, 264, 265, 266, 267, 268, 269, 69, 270, 271, 272, 273, 274, 44, 275, 276, 277, 278, 279, 280, 281, 282, 283, 284, 285, 286, 287, 57, 288, 289, 290, 291, 292, 293, 120, 294, 73, 295, 49, 296, 297, 298, 299, 300, 301, 302, 303, 304, 305, 306, 84, 307, 308, 309, 310, 311, 61, 70, 312, 313, 71, 314, 315, 316, 317, 41, 43, 318, 319, 320, 321, 322, 323, 324, 325, 326, 327, 328, 329, 31, 330, 331, 332, 333, 66, 334, 335, 27, 59, 336, 60, 337, 338, 33, 76, 72, 115, 339, 340, 341, 74, 342, 113, 343, 344, 345, 50, 346, 347, 348, 349, 350, 351, 45, 47, 352, 353, 354, 355, 356, 357, 358, 62, 359, 360, 361, 362, 363, 54, 364, 365, 53, 366, 40, 367, 368, 369, 370, 371, 372, 373, 374, 375, 376, 377, 378, 379, 380, 381, 382, 383, 384, 385, 386, 387, 56, 55, 388, 389, 390, 391, 392, 393, 394, 395, 396, 397, 398, 399, 400, 401, 402, 403, 404, 405, 406, 407, 408, 409, 410, 42, 411, 77, 1, 412, 413, 414, 415, 416, 417, 418, 419, 420, 421, 64, 63, 422, 423, 424, 425, 67, 68, 426, 427, 428, 58]

- **Others** [429, 28, 430, 16, 5, 26, 431, 432, 433, 434, 435, 436, 437, 438, 439, 440, 441, 442, 101, 51, 82, 132, 443, 131, 129, 103, 105, 65, 444, 445, 446, 447, 98, 100, 18, 127, 102, 130]

## 10 Alignment Goals and Human Values

### 10.1 A Comprehensive Taxonomy of Human Values

This conventional theory was developed without the context of human-AI interaction, which might overlook values that need to be considered for human-AI alignment. Therefore, we used a *bottom-up* approach to extract all values studied in our collected alignment literature, mapped them onto the

| | Goals | Definitions | Limitations / Risks |
|---|---|---|---|
| **The Goal of Alignment** | **Instructions** | The agent does what I instruct it to do. | On a larger scale, it is difficult to precisely specify a broad objective that captures everything we care about, so in practice the agent will probably optimise for some proxy that is not completely aligned with our goal. |
| | **Intentions** or (Expressed Intentions) | The agent does what I intend it to do. | It is quite possible for intentions to be irrational or misinformed, or for the principal to form an intention to do harmful or unethical things. |
| | **Preferences** or (Revealed Preferences) | The agent does what my behaviour reveals I prefer. | 1) People have preferences for things that harm them. 2) People have preferences about the conduct of other people. 3) Preferences are not a reliable guide to what people really want or deserve due to adaptiveness. |
| | **Desires** or (Informed Preferences) | The agent does what I would want it to do if I were rational and informed. | Researchers would have to apply a corrective lens or filter to the preferences they actually observe. As a consequence, the approach is no longer strictly empiricist. |
| | **Interest** or (Well-being) | The agent does what is in my interest, or what is best for me, objectively speaking. | Something in a human's interest does not mean he/she ought to do it or is morally entitled to do so, such as an interest in stealing. Also, it is hard to manage trade-offs the collective interests of different people. |
| | **Values** | The agent does what it morally ought to do, as defined by the individual or society. | Current the best possibility, but it still encounters two difficulties of 1) specifying what values or principles, and 2) concerning the body of people who select the principles with which AI aligns. |

Table 2: The **Goals** of Alignment. We present the six prevailing alignment goals, associating with their Definitions (middle column), Limitations and Risks (right column). We consider **Human Values** as the main goal of alignment in this work referring to an extensive analysis and arguments in existing studies [29, 30]

Schwartz Theory of Basic Values, and supplemented the theory with AI-related structure and content. As a result, we identified the structural relationships among human values and mapped existing literature to a fine-grained taxonomy (see Table 3). We supplemented the traditional theory's four high-order value types (*i.e.,* "Self-Enhancement", "Openness to Change", "Conservation", "Self-Transcendence") with a novel high-order value type, named "Desired Values for AI Tools" that encompasses two motivational value types (*i.e.,* "Usability" and "Human-Likeness"). We further organize the relationship among these value types along two dimensions [39]: different resources (*i.e.,* individuals, society and interaction) and different self-intentions (*i.e.,* self-protection against threat and self-expansion and growth). Furthermore, we elaborate the definitions of the 12 motivational value types and their exemplary values by mapping them to relevant human-AI alignment papers from our corpus in Table 3. During the process of mapping, we found: 1) value terms in empirical papers were often named differently (*e.g.,* capability and competence), or check opposites (*e.g.,* fairness and bias); 2) there are many values not studied in our corpus, *i.e.,* indicated as (•) in the Figure.

## 10.2 Insights into Human Values for Alignment

Our analysis, based on the adaptation of Schwartz's Theory of Basic Values and our comprehensive literature review, identifies three critical findings for future research:

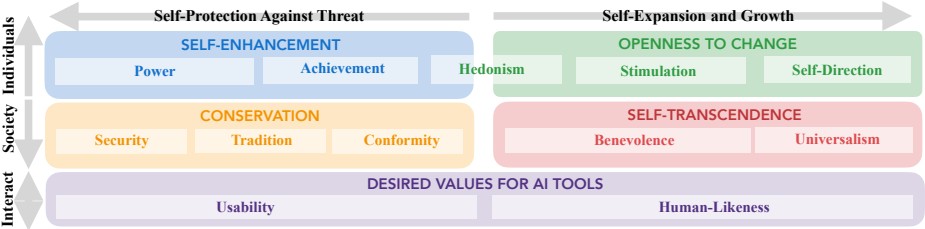

Table 3: The value relations and taxonomy. We consider 5 high-order value types encompassing 12 motivational value types, indicated by their sources (e.g., individuals, society and interaction).

**Value Prioritization in AI Systems.** Human value systems are not merely subsets of values, but ordered systems with relative priorities [38, 39]. For instance, [39] presented the definition for this phenomenon: *"a value is ordered by importance relative to other values to form a system of value priorities. The relative importance of multiple values guides action....The trade-off among relevant, competing values guides attitudes and behaviors."*. Current AI alignment algorithms, often based on datasets of human preferences [1, 287, 58], may inadvertently prioritize majority values, potentially neglecting those of marginalized groups [349]. Future research should address this complex interplay of values in AI systems.

**Universal vs. Personalized AI Values**. While certain values are universally expected from AI (e.g., capability, equity, responsibility), others may be undesirable in specific contexts [33] (e.g., seeking power). Simultaneously, AI models should be adaptable to diverse human value systems [28]. Research is needed to develop methods for identifying appropriate value sets for specific individuals or groups, and for customizing AI to align with user values while maintaining ethical principles.

**Disparities in Value Expectations and Evaluation**. The fundamental differences between humans and AI necessitate distinct approaches to value evaluation. For instance, assessing AI honesty may require mechanistic interpretability [448], a more rigorous standard than that applied to humans. Future studies should explore methods for evaluating and explaining AI values and calibrating human expectations accordingly.

## 11   Interaction Techniques for Specifying Human Values

Our research reveals disparities in interaction techniques for human-AI value alignment across AI-centered (NLP/ML) and human-centered (HCI) domains. As depicted in Figure 4, this analysis focuses on three key areas:

**Domain-Specific Interaction Techniques**. The interaction techniques in AI-centered (NLP/ML) and Human-centered (HCI) alignment studies are often differ [449]. NLP/ML studies primarily utilize numeric and natural language-based techniques. Also, NLP/ML research explore implicit feedback to extract human hidden feedback. In contrast, HCI research encompasses a broader range of graphical and multi-modal interaction signals (e.g., sketches, location information) beyond text and images. This disparity suggests potential gaps in extracting comprehensive human behavioral information.

**Stage-Specific Interaction Techniques**. In NLP/ML, the learning stage predominantly employs rating and ranking interactions for alignment in dataset generation. However, when humans use AI in the inference stage, as demonstrated in HCI research, involves more diverse user interactions. This discrepancy highlights the need for alignment between model development and practical deployment.

**Divergent Data Utilization**. NLP/ML typically uses interaction outputs as training datasets, while HCI analyzes this data to understand human behavior and feedback. As AI systems evolve, developing new interaction modes to capture a broader spectrum of human expression becomes crucial.

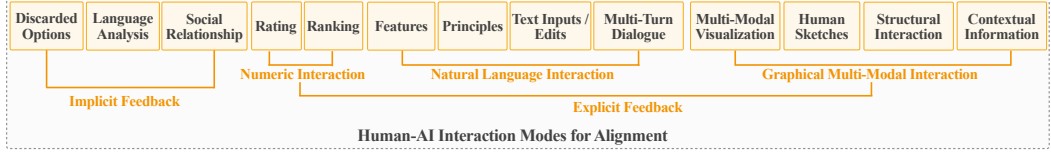

Figure 4: The interaction techniques for specifying values in human-AI alignment.

> **Takeaways of Interaction Techniques for Alignment.**
> 1. Some common human feedback styles used in NLP/ML are not often studied in HCI.
> 2. Diverse human interactive feedback in HCI are not fully used in AI development in NLP/ML.

## 12   Challenges in Achieving Alignment

The concept of *alignment* in AI research has a long history, tracing back to 1960, when AI pioneer Norbert Wiener [450] described the AI alignment problem as: "*If we use, to achieve our purposes, a mechanical agency with whose operation we cannot interfere effectively ... we had better be quite*

*sure that the purpose put into the machine is the purpose which we really desire.*" Discussion around intelligent agents and the associated concerns relating to ethics and society have emerged since then [451]. Next, we discuss the well-known challenges encountered in achieving alignment.

**Challenge 1: Outer and Inner Alignment.** In the context of "intelligent agents," until now, *AI alignment* research has aimed to ensure that any AI systems that would be set free to make decisions on our behalf would act appropriately and reduce unintended consequences [452, 451, 453]. At the *near-term* stage, aligning AI involves two main challenges: carefully *specifying* the purpose of the system (*outer alignment i.e.,* providing well-specified rewards [33]) and ensuring that the system adopts the specification robustly (*inner alignment, i.e.,* ensuring that every action given an agent in a particular state learns desirable internally-represented goals [33]). Significant efforts have been made, for *inner alignment*, to align AI systems to follow alignment goals of an individual or a group (*e.g.,* instructions, preferences, values, and/or ethical principles) [1] and to evaluate the performance of alignment [16]. However, for *outer alignment*, AI designers are still facing difficulties in specifying the full range of desired and undesired alignment goals of humans.

**Challenge 2: Specification Gaming.** To learn human alignment goals, AI designers typically provide an objective function, instructions, reward function, or feedback to the system, which is often unable to completely specify all important values and constraints that a human intended [6]. Hence, AI designers resort to easy-to-specify proxy goals such as *maximizing the approval of human overseers* [4], which results in "*specification gaming*" [7] or "*reward hacking*" [8] issues (*i.e.,* AI systems can find loopholes that help them accomplish the specific objective efficiently but in unintended, possibly harmful ways). Additionally, the black-box nature of neural networks brings additional ethical and safety concerns for alignment because humans don't know about the inner state and the actions AI leveraged to achieve the output. Consequently, AI systems might make "correct" decisions with "incorrect" reasons, which are difficult to discern. Society is already facing these issues, such as data privacy [9], algorithmic bias [10], self-driving car accidents [11], and more. As a result, these considerations necessitate considering human-AI interaction in AI alignment for specification and evaluation, ranging from addressing problems around who uses an AI system, with what goals to specify, and if the AI system perform its intended function from the user's perspective.

**Challenge 3: Scalable Oversight.** From a long-term perspective, when advanced AI systems become more complex and capable (*e.g.,* AGI [12]), it becomes increasingly difficult to align them to human values through human feedback. Evaluating complex AI behaviors applied to increasingly challenging tasks can be slow or infeasible for humans to ensure all sub-steps are aligned with their values [5]. Therefore, researchers have begun to investigate how to reduce the time and effort for human supervision, and how to assist human supervisors, referred to as *Scalable Oversight* [13].

**Challenge 4: Dynamic Nature.** As AI systems become increasingly powerful, the alignment solutions must also adapt dynamically since human values and preferences change as well. As [14] posit, AI systems may be neither humane nor desirable if we do not ask questions about the long-term cognitive and social effects of social agent systems (*e.g., how will agent technology affect human cognition*). All these considerations call for a long-term and dynamic perspective to address human-AI alignment as an ongoing, mutual process with the collective efforts of cross-domain expertise.

**Challenge 5: Existential Risk.** Further, some AI researchers claim that [454] advanced AI systems will begin to seek power over their environment (*e.g.,* humans) once deployed in real-world settings, as such behavior may not be noticed during training. For example, some language models seek power in text-based social environments by gaining money, resources, or social influence [455]. Consequently, some hypothesize that future AI, if not properly aligned with human values, could pose an *existential risk* to humans [456].

# 13   Author contributions

This project was a team effort, built on countless contributions from everyone involved. To acknowledge individual authors' contributions and enable future inquiries to be directed appropriately, we followed the ACM's policy on authorship [457] and listed contributors for each part of the paper.

## 13.1   Overall Author List and Contributions

**Project Lead**

The project lead initialized and organized the project, coordinated with all authors, participated in the entire manuscript.

- **Hua Shen (NYU Shanghai, New York University, huashen@nyu.edu)**: Initiated and led the overall project, prepared weekly project meetings, filtered papers, designed dimensions and codes (initial, revision), coded all papers, initiated the framework and developed human value and interaction modes analysis figures, participated in drafting all sections, paper revision and polishing.

**Team Leads**

The team leads organized all team events, coordinated with leads and members, contributed to a portion of manuscript.

- **Tiffany Knearem (MBZUAI, tiffany.knearem@mbzuai.ae.ac)**: Led the HCI team, prepared weekly team meetings, filtered papers, designed dimensions and codes (initial, revision), coded partial papers, ideated the framework and analysis and future work content, participated in writing (Critical Thinking and AI Impact on Human sections), paper revision and polishing.
- **Reshmi Ghosh (Microsoft, reshmighosh@microsoft.com)**: Led the NLP/AI team, prepared weekly team meetings, filtered papers, coded partial papers, ideated the framework and analysis and future work content, participated in writing (AI evaluation section), paper revision and polishing.

**Team Members (Alphabetical)**

The team members contributed to a portion of paper review, regular discussions, and drafted a portion of the manuscript.

- **Kenan Alkiek (University of Michigan, kalkiek@umich.edu)**: filtered papers, coded partial papers, data processing and analysis, ideated paper analysis and future work, paper revision and polishing, mainly involved in NLP Team
- **Kundan Krishna (Apple, kundank@andrew.cmu.edu)**: filtered papers, coded partial papers, ideated the framework and future work, participated in writing (Customizing AI section), designed dimensions and codes (initial, revision), paper revision and polishing, mainly involved in NLP Team
- **Yachuan Liu (University of Michigan, yachuan@umich.edu)**: filtered papers, coded partial papers, participated in writing (revised Integrate General Value and Customization content sections), paper revision and polishing, mainly involved in NLP Team
- **Ziqiao Ma (University of Michigan, marstin@umich.edu)**: filtered papers, coded partial papers, designed dimensions and codes (initial, revision), developed Human Value category, participated in writing (Human Value taxonomy, revised representation, and value gap analysis sections), paper revision and polishing, mainly involved in NLP Team
- **Savvas Petridis (Google PAIR, petridis@google.com)**: filtered papers, coded partial papers, ideated the interaction-related analysis and future work, participated in writing (Perceive and Understand AI), paper revision and polishing, mainly involved in HCI Team
- **Yi-Hao Peng (Carnegie Mellon University, yihaop@cs.cmu.edu)**: filtered papers, coded partial papers, participated in writing (Human-AI Collaboration section), paper revision and polishing, mainly involved in HCI Team
- **Li Qiwei (University of Michigan, rrll@umich.edu)**: filtered papers, coded partial papers, ideated the interaction-related taxonomy and analysis, participated in writing (Interaction Mode section), mainly involved in HCI Team
- **Sushrita Rakshit (University of Michigan, sushrita@umich.edu)**: filtered papers, coded partial papers, participated in writing (Integrate General Value section), paper revision and polishing, mainly involved in NLP and HCI Team
- **Chenglei Si (Stanford University, clsi@stanford.edu)**: filtered papers, coded partial papers, designed dimensions and codes (initial, revision), ideated the framework and future

work, participated in writing (Assessment of Collaboration and Impact section), paper revision and polishing, mainly involved in HCI Team

- **Yutong Xie (University of Michigan, yutxie@umich.edu)**: filtered papers, coded partial papers, designed dimensions and codes (initial, revision), ideated the value representation taxonomy, participated in writing (Human Value Representation section), paper revision and polishing, , mainly involved in NLP Team

## Advisors (Alphabetical)

The advisors involved in and made intellectual contributions to essential components of the project and manuscript.

- **Jeffrey P. Bigham (Carnegie Mellon University, jbigham@cs.cmu.edu)**: contributed to the framework on aligning human to AI direction, vision on the status quo of alignment research, and future work discussions, and participated in paper revision and proofreading.

- **Frank Bentley (Google, fbentley@google.com)**: contributed to the historical context and project objectives, improved the definitions and design of research methodology, and participated in paper revision and proofreading.

- **Joyce Chai (University of Michigan, chaijy@umich.edu)**: iteratively involved in developing and revising definitions and the framework on aligning AI to human direction, advised on analysis and future work, and participated in paper revision and proofreading.

- **Zachary Lipton (Carnegie Mellon University, zlipton@cmu.edu)**: contributed insights from Machine Learning, NLP, and AI fields to revise the definitions and framework on aligning AI to human direction, and participated in paper revision and proofreading.

- **Qiaozhu Mei (University of Michigan, qmei@umich.edu)**: contributed insights from Data Science, Machine Learning, and NLP fields to improve definitions and the framework on aligning AI to human direction, and participated in paper revision and proofreading.

- **Rada Mihalcea (University of Michigan, mihalcea@umich.edu)**: involved in framing and revising the structure and taxonomy of human values, and contributed to improving the manuscript's title, introduction, and other sections, and participated in paper revision and proofreading.

- **Michael Terry (Google Research, michaelterry@google.com)**: contributed arguments and vision on the status quo of alignment research, framed project objectives and contributions, improved definitions and data analysis, and participated in paper revision and proofreading.

- **Diyi Yang (Stanford University, diyiy@stanford.edu)**: involved in improving definitions and the framework, contributed social insights to the work, and participated in paper revision and proofreading.

## Project Leading Advisors

The project leading advisors actively involved in the entire project process and all manuscript sections.

- **Meredith Ringel Morris (Google DeepMind, merrie@google.com)**: iteratively involved in drafting all sections, contributed to core argument ideation, framework and definition improvement, provided future work insights, and participated in paper drafting, revision, and proofreading on all sections.

- **Paul Resnick (University of Michigan, presnick@umich.edu)**: actively involved and advised on the entire project process, including initiating the project and research agenda, iteratively improved definitions, framework, and analysis, and participated in paper revision and proofreading.

- **David Jurgens (University of Michigan, jurgens@umich.edu)**: provided advice throughout the project, including iterative discussions on project milestones and content ideation, organized several meetings to receive feedback from external audiences, and participated in paper revision and proofreading.

## ACKNOWLEDGMENTS

We thank Eric Gilbert for his constructive feedback on human-centered insights on alignment, thank Eytan Adar for his valuable guidance on designing the interaction techniques for human-AI alignment, and thank Elizabeth F. Churchill for her insightful discussion on this manuscript. We also thank Michael S Bernstein, Denny Zhou, Cliff Lampe, and Nicole Ellison for their encouraging feedback on this work. We welcome researchers' constructive discussions and interdisciplinary efforts to achieve long-term and dynamic human-AI alignment collaboratively in the future. This work was supported in part by the National Science Foundation under Grant No. IIS-2143529 and No. IIS-1949634.

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
