# OpenReview forum: "Position: Towards Bidirectional Human-AI Alignment"
_NeurIPS.cc/2025/Position_Paper_Track — NeurIPS 2025 Position Paper Track_

### Official Review · Reviewer_fjCn · 2025-08-09

**Significance:** 4
**Presentation:** 4
**Rating:** 10
**Confidence:** 5

**Summary:**

This position paper conducts a systematic literature review of 411 papers in the AI alignment literature. It then introduces the "Bidirectional Human-AI Alignment Framework," which consists of 1) aligning AI values to human values and 2) aligning humans to AI (i.e., enabling humans to adapt to rapid AI development and the associated social, political, economic changes). Through qualitative coding of the papers included in the systematic review, the paper then identifies gaps in the alignment literature such as human-in-the-loop evaluation, fostering AI literacy, human-AI collaborative dynamics, and ethical auditing.

The paper then outlines three important directions for future research:
- Specification Game: how can human values be fully specified and then integrated into AI systems? How can human values be elicited from dynamic interfaces beyond instructions, rankings, and ratings?
- Dynamic Co-evolution of Alignment: how can AI systems keep up with/update to reflect evolving human values? How do AI systems themselves influence human values? How can humans adapt to rapidly advancing AI capabilities?
- Safeguarding Co-adaptation: how can values in AI systems be made interpretable, and how can we intervene on those values?

**Strengths:**

- [Major] Support: this position paper is based on a systematic literature review of over 400 papers. Reviewed papers are individually tagged/categorized in order to determine which areas of alignment research are most/least covered, which is useful quantitative evidence for the position. The tags/categories also appear comprehensive.
- [Major] Significance & context: the paper sufficiently demonstrates the importance of bidirectional alignment, and the three directions for future research are well-justified, cited, & explained. I appreciated how specific some of the research ideas were, which made the paper very actionable and also useful given the number of citations to prior work (e.g., the discussion in lines 316-30)
- [Major] Discussion potential: the target audience is the research community, and the paper makes a strong case for the importance of the research directions it outlines. As a result, I would expect the paper to inspire significant discussion within the community.
- [Major] Communication quality: the paper is very well written, and the tables and figures are very informative. I thought the definitions in Section 2 were very clear, and I also appreciated Figures 2 and 3 and the examples given of each category.

**Weaknesses:**

- [Minor] Methodology: the paper should be more transparent about its methodology
    - Although the systematic review consisted of 411 papers, these 411 papers are not listed or cited anywhere (given that there are only 143 citations). I would highly recommend citing all reviewed papers, then adding a table listing all of them
    - Line 882 indicates that the systematic review "adher[es] to the the [sic] PRISMA guideline." PRISMA is a guideline for transparent _reporting_ of systematic literature reviews, but the paper does not actually follow PRISMA reporting guidelines. Highly suggest adding an appendix with the PRISMA checklist [1-2]
    - In particular, it would be useful to know: how the initial 34,213 papers were identified; what the inclusion/exclusion criteria were for filtering; how the codebook was developed; how coding was performed/how many coders did each paper/how conflicts were resolved, etc.
- [Minor] Context: lines 331-404 could use more citations—in particular, I would like to see citations to work from other disciplines or in non-AI contexts that are good examples of the type of work that you'd like to see (if such work exists)

[1] https://www.prisma-statement.org/

[2] https://prisma.shinyapps.io/checklist/

**Questions:**

- Lines 44-45: suggest rephrasing to active voice, i.e., "the research community should/must..."
- L95-100: consider moving this definition to the top of Section 2, which might make the definitions easier to follow
-L 411-12: how did you filter for ML, NLP, and HCI only? Can you list some examples of papers from other disciplines that are relevant?
- Framing: "align AI to humans" refers to aligning AI and human _values_, but "align humans to AI" refers to human "human cognitive and behavioral adaptation to AI advancement" (L105-6). Since these are different "axes" that are being aligned, I find the "align humans to AI" language a bit confusing since my instinct is to read that statement as "align human values to AI values" (which is incorrect). Suggest changing the language, maybe "align AI to human values" and "align humans to AI capabilities"?
- Captions of figures/tables: is "topology" meant to be "typology"?
- Consider adding citations to [3-4] and maybe the AI resilience literature [5-6]

[3] https://www.full-stack-alignment.ai/paper

[4] https://arxiv.org/abs/2405.10295

[5] https://cetas.turing.ac.uk/sites/default/files/2023-08/cetas-cltr_ai_risk_briefing_paper.pdf

[6] https://thefuturesociety.org/aicrisisexplainer/

**Alternative Position:**

No

**Author Identification:**

No.

**Context:**

4

**Discussion:**

4

**Ethics:**

["NO or VERY MINOR ethics concerns only"]

**Position:**

Yes, the paper argues for or against a position related to machine learning.

**Support:**

4

**Thoroughness:**

4

---

### Official Review · Reviewer_Pq5E · 2025-08-11

**Significance:** 3
**Presentation:** 2
**Rating:** 8
**Confidence:** 5

**Summary:**

This position paper introduces a "Bidirectional Human-AI Alignment" framework that extends traditional AI alignment from a one-way process of aligning AI to humans into a reciprocal relationship where humans also adapt to AI systems. Through a systematic review of over 400 papers spanning HCI, NLP, and ML, the authors organize this framework around four research questions:  (RQ1) What relevant human values are studied for AI alignment, and how do humans specify these values? (RQ2) How can human values be integrated into the AI systems? (RQ3) How do humans learn to perceive, explain, and critique AI? (RQ4) How do individuals and society adapt their behaviours in response to AI advancements? The paper concludes with three progressive challenges: the "Specification Game" requiring better methods to capture human values through democratic processes; "Dynamic Co-evolution" necessitating continuous adaptation where AI systems and humans evolve together; and "Safeguarding Co-adaptation" demanding interpretable AI architectures with robust oversight to prevent harmful autonomous actions, ultimately arguing for alignment as an ongoing, mutual adaptation process.

**Strengths:**

* One area of strength lies in its successful unification of disparate research communities and perspectives under a coherent bidirectional framework. It excellently bridges the significant gap between ML/NLP and HCI communities, which have been working on different aspects of alignment in isolation - some focused on aligning AI to human values, others on human adaptation to AI systems.
* The framework systematically integrates these previously siloed research streams while thoughtfully incorporating the critical but often overlooked temporal dimension of alignment, recognizing that both human values and AI capabilities evolve dynamically over time.
* The authors provide clear, standardized terminology that helps unify fragmented field vocabulary, supported by a robust systematic review of 400+ papers that lends empirical credibility to their gap identification and claims.
* Also, the paper demonstrates that alignment isn't merely a technical problem but requires a holistic approach considering human cognitive adaptation, behavioral changes, and broader societal impacts, effectively articulating current research gaps, underexplored dimensions, and potential future challenges.

**Weaknesses:**

* The paper suffers from some structural and evidential limitations that undermine its impact. A major flaw is the heavy reliance on appendix materials, where essential content, including comprehensive value taxonomies, definitional frameworks, and systematic methodology, is buried rather than integrated into the main narrative, creating a fragmented reading experience where the main text cannot stand alone.

* The paper lacks some real-world scenarios demonstrating the urgent risks of maintaining current unidirectional alignment approaches, without concrete examples of potential failures or misalignment consequences.

* Despite reviewing 400+ papers systematically, the main text lacks sufficient quantitative analysis and relies too heavily on qualitative descriptions. The main text/body of the paper misses opportunities to provide statistical evidence through concrete metrics on research distributions, citation patterns, and temporal trends, particularly weakening claims about ML/NLP vs. HCI disparities that would benefit from numerical data rather than general assertions.

The content is valuable. I'd suggest restructuring the paper in the final version.

**Questions:**

* How would you mathematically formalize bidirectional optimization that simultaneously trains AI systems and adapts human behavior, rather than current single-direction loss functions? How would you make it even more dynamic to cover the evolution of values?
* What algorithmic methods would validate that training data represents authentic human values before assuming it's a "gold standard"?
* When AI capability goals conflict with human agency preservation, what mathematical frameworks would resolve these trade-offs?
* What specific legal requirements would operationalize bidirectional alignment (e.g., mandatory human adaptation impact assessments)?
* How would you structure enforceable agreements between AI companies, users, and regulators across different jurisdictions?
* How would bidirectional optimization scale computationally for systems like ChatGPT or Claude serving millions of diverse users?
* With conflicting user values and adaptation needs, how would your framework handle this complexity while maintaining system safety?

**Alternative Position:**

Yes, and alternative positions are well-considered and addressed by the argument

**Author Identification:**

No.

**Context:**

3

**Discussion:**

4

**Ethics:**

["NO or VERY MINOR ethics concerns only"]

**Position:**

Yes, the paper argues for or against a position related to machine learning.

**Support:**

2

**Thoroughness:**

4

---

### Official Review · Reviewer_9e5Q · 2025-08-11

**Significance:** 2
**Presentation:** 2
**Rating:** 6
**Confidence:** 4

**Summary:**

The paper reviews over 400 works across HCI, NLP, ML, and related areas to examine how “AI alignment” is defined and practiced. It proposes the Bidirectional Human-AI Alignment framework, which pairs aligning AI to human values with aligning humans to AI through cognitive, behavioral, and societal adaptation. The study identifies gaps in long-term interaction design, human value modeling, and mutual understanding. It also outlines three core challenges specification gaming, scalable oversight, and dynamic alignment and offers recommendations for more reciprocal and adaptive alignment approaches.

**Strengths:**

The paper clearly articulates its central position that alignment should be understood as a bidirectional, dynamic process, expanding beyond the traditional one-way perspective.
It supports its argument with a large-scale systematic review of over 400 papers, giving its claims breadth and grounding in cross-disciplinary literature.
The Bidirectional Human-AI Alignment framework is well-structured and effectively links conceptual ideas to practical dimensions, making the proposal tangible.

**Weaknesses:**

Although it identifies major research gaps, it offers few concrete methodologies or metrics for implementing and measuring “aligning humans to AI” in practice.

**Questions:**

How might the framework be stress-tested in real-world, high-stakes environments to ensure its applicability beyond academic or prototype settings?

Have you considered whether certain AI capabilities or domains (e.g., medical AI vs. creative AI) require different balance points between the two alignment directions?

**Alternative Position:**

Yes, and alternative positions are well-considered and addressed by the argument

**Author Identification:**

No.

**Context:**

2

**Discussion:**

3

**Ethics:**

["NO or VERY MINOR ethics concerns only"]

**Position:**

Yes, the paper argues for or against a position related to machine learning.

**Support:**

2

**Thoroughness:**

5

---

### Note · Authors · 2025-08-28

**1-10 Additional Comments:**

None

**1-11 Submit Again:**

Definitely yes

**1-1 Submission Process:**

4

**1-2 Next Year:**

It would be helpful to provide a clearer timeline for the position track, enable longer reaction length to address reviewers' questions, or to include an author rebuttal stage to address potential reviewer misunderstandings or concerns.

**1-3 Future Development:**

The current NeurIPS Position Paper track is overall excellent, with only a few minor areas for improvement noted above.

**1-4 Interest:**

["Panel discussions with other position paper authors", "Structured debates on controversial topics", "Workshops for developing position papers", "Mentorship programs for early-career researchers"]

**1-4 Other Interest:**

None

**1-5 Thoughtful:**

6

**1-6 Supportive:**

9

**1-7 Technical Aspects Versus Position:**

3

**1-8 Gate Keeping:**

10

**1-9 Camera Ready Changes:**

We sincerely thank all reviewers for their constructive and encouraging feedback. Reviewer-9e5Q valued our framing of alignment as a bidirectional process and noted the framework is “well-structured and effectively linking conceptual ideas to practical dimensions.” Reviewer-Pq5E highlighted the paper as a “successful unification of disparate research communities and perspectives under a coherent bidirectional framework,” praising the “clear, standardized terminology” and holistic view beyond technical fixes. Reviewer-fjCn recognized that our work “makes a strong case for the importance of the research directions,” is “very actionable,” and provides “useful quantitative evidence” through systematic categorization of over 400 papers.

We also appreciate their constructive suggestions, which we will incorporate to improve clarity, rigor, and practical relevance. Key planned revisions include:

1. Methodological Transparency. Expand the main text with review process details, including inclusion/exclusion criteria, codebook, a PRISMA flow diagram, and explicit reference to all 411 papers.

2. Integration of Key Content. Move core taxonomies and quantitative results from the appendix into the main text to make the paper self-contained.

3. Concrete Case Studies. Add domain-specific examples (e.g., medical vs. creative AI) and cross-disciplinary illustrations beyond ML/NLP/HCI to highlight bidirectional alignment importance and potential failure modes of unidirectional approaches.

4. Clarifying Terminology and Visuals. Refine terms to prevent misunderstanding, update figure/table captions to “typology,” and revise section phrasing for clarity.

5. Broader Perspectives and Future Directions. Add cross-disciplinary citations (human factors, cognitive science, AI resilience) and expand discussions on formalizing bidirectional alignment, validating value representations, handling trade-offs, and scaling alignment for diverse users.

**3-1 Review Response1:**

9e5Q

**3-2 Reaction To Review1:**

We thank the reviewer for the valuable feedback, especially the support for understanding alignment as a bidirectional process and recognizing our framework as “well-structured and effectively linking conceptual ideas to practical dimensions.” We address the concerns below:

W- Methodologies and Metrics for “Aligning Humans to AI”. Beyond the conceptual stance (lines 95–100), we also outline methodologies and metrics in RQ3–RQ4 (lines 199–272).
- Methodologies: Our Evaluation in Human Studies section (lines 258–267) highlights practical methods, including interaction analytics, interviews, behavioral analyses, and societal-scale approaches such as opinion surveys.
- Metrics: Measures such as task success, efficiency, and satisfaction assess how effectively humans adapt to AI (lines 259–262).
Our position paper does not aim to develop concrete implementations but to overview existing methods and identify gaps and future directions.

Q1- Stress-Testing in High-Stakes Environments. We agree that real-world validation is essential. Our blueprint includes (i) defining success/failure criteria, (ii) staged rollout with exposure to edge cases and adversarial inputs, and (iii) continuous monitoring via metrics and feedback. Stressors may be introduced through red-teaming, fault testing, and “fire drills,” with safeguards ensuring accountability. While detailed protocols are beyond scope, we will add this as an Appendix example.

Q2- Domain-Specific Balance Points. Our framework is generalizable (lines 51–70), but balance varies: safety-critical fields (e.g., medicine) emphasize aligning AI to human values, while creative/collaborative domains may prioritize human adaptation to AI. Recent work [1] highlights the need for scenario-specific alignment. We will clarify that while our framework offers shared vocabulary, balance should be tuned to domain-specific risks and goals.

[1] Mind the Value-Action Gap: Do LLMs Act in Alignment with Their Values? EMNLP 2025.

**3-3 Review Response2:**

Pq5E

**3-4 Reaction To Review2:**

We thank the reviewer for the thoughtful and constructive feedback, and for recognizing our work as a “successful unification of disparate research communities and perspectives under a coherent bidirectional framework,” with “clear, standardized terminology” and a holistic view of alignment beyond technical fixes. We will restructure the paper to better integrate these valuable points：

W1 – We agree key definitions, taxonomies, and methods should appear in the main text. Sec. 2 will be expanded with summaries of core taxonomy, frameworks, and methodology now in the appendix, ensuring the paper stands alone.

W2 – We will add concrete case studies (e.g., automation bias in clinical AI) to illustrate risks of unidirectional alignment and emphasize the urgency of addressing human adaptation failures.

W3 – Quantitative analysis (Fig. 4, Secs. 4.1–4.2) will be moved into the main text with numeric highlights and collection dates, providing transparent statistical evidence despite the field’s rapid evolution.

Q1 – On formalizing bidirectional optimization: while our focus is framing and motivation, we will sketch directions treating alignment as a multi-objective optimization problem, with game theory and multi-agent RL as promising tools.

Q2- For validating value representations, we propose methods such as Controlled Value Confounding Tests, introduced as an open research challenge.

Q3- To address trade-offs between capability and agency, we frame agency-preservation as a constraint, supported by preference-learning, social-choice aggregation, corrigibility, and multi-value heads.

Q4 & 5 – Though not our focus, we recognize regulatory and governance importance. We will note directions such as Human-Adaptation Impact Assessments and multi-layer governance stacks.

Q6 & 7 – Scaling requires societal-level measurement and adaptive mechanisms, supported by our framework’s pluralistic alignment, social-choice aggregation, and oversight for transparency and safety.

**3-5 Review Response3:**

fjCn

**3-6 Reaction To Review3:**

We sincerely thank the reviewer for the generous and encouraging feedback, and for recognizing our paper as “making a strong case for the importance of the research directions,” “very actionable,” and providing “useful quantitative evidence” through the systematic categorization of over 400 papers. We will incorporate the constructive suggestions below into the final version.

W1 – Methodology transparency and PRISMA. We agree on the need for greater transparency. The revision will add an Appendix with: (1) a full list of reviewed papers; (2) a PRISMA flow diagram and checklist; and (3) details of the review process, such as venues searched, criteria, codebook, and more. The full list will also be made available on a companion website.

W2 – Cross-disciplinary citations. We will strengthen interdisciplinary grounding by adding references from psychology, cognitive and social sciences -- adding literatures that parallel human adaptation to AI. For example, Dautenhahn (2000) [7] highlights co-evolution of technology and cognition, and Taylor (1983) [8] analyzes cognitive adaptation to disruption.

Q1-Q4 – Clarifications and rewording. We will: (1) use more active phrasing; (2) move the key definition to the start of Section 2; (3) clarify terminology distinguishing “aligning AI to human values” from “aligning humans to AI capabilities”; (4) update figure/table captions to “typology”; and (5) add examples beyond ML/NLP/HCI, including Hofstede’s Values Survey Module (1984) [9], Thorndike & Stein’s social intelligence (1937) [10], and more.

Q5 – Additional citations. We will incorporate the suggested works on alignment, societal adaptation [3-4], and AI resilience [5-6], enriching the discussion with broader disciplinary perspectives.


[7] https://www.torrossa.com/en/resources/an/5002536
[8] https://psycnet.apa.org/record/1984-17824-001
[9] https://journals.sagepub.com/doi/abs/10.1177/017084068400500423
[10] https://psycnet.apa.org/record/1937-03825-001

---

### Meta-Review · Area_Chair_nJgi · 2025-08-31

**Rating:** 9
**Confidence:** 5

**Strengths:**

* All reviewer agree that the position is interesting, timely and likely spark discussion.
* The opinion is clearly argued.
* The piece is clearly relevant.

**Weaknesses:**

The reviewers point to a list of elements useful for improving the paper. These are all minor improvements, especially addressing elements that could go in the appendix (like the list of all papers reviewed).
One major need for improvement is an explanation of the selection criteria of the papers (see the discussion between the reviewers on this point).

**Questions:**

Based on the reviewers commens I suggest think about the following questions
* How can we develop a mathematical and computational framework for bidirectional alignment that is scalable, handles conflicting values, and validates the authenticity of human data?
* What legal and regulatory structures are needed to operationalize and enforce bidirectional alignment across different jurisdictions?
* How can the paper's terminology, structure, and citations be improved to make the arguments clearer and more precise?
* How could you properly acknowledge the vast amount of literature in the social science addressing human values and the difficulty of measuring them?
* How do you envision value change over time and across locations to be implemented on a regular basis and fair?

**Ethics:**

No ethics concerns raised.

**Thoroughness:**

3

---

### Decision · Program_Chairs · 2025-09-26

Accept